# Salvianolic Acid B in Microemulsion Formulation Provided Sufficient Hydration for Dry Skin and Ameliorated the Severity of Imiquimod-induced Psoriasis-like Dermatitis in Mice

**DOI:** 10.3390/pharmaceutics12050457

**Published:** 2020-05-17

**Authors:** Jiun-Wen Guo, Yu-Pin Cheng, Chih-Yi Liu, Haw-Yueh Thong, Chi-Jung Huang, Yang Lo, Chen-Yu Wu, Shiou-Hwa Jee

**Affiliations:** 1Department of Medical Research, Cathay General Hospital, Taipei 10630, Taiwan or cgh393831@cgh.org.tw (J.-W.G.); or aaronhuang@cgh.org.tw (C.-J.H.); 2Program in Pharmaceutical Biotechnology, College of Medicine, Fu Jen Catholic University, New Taipei City 24205, Taiwan; 3Department of Dermatology, Cathay General Hospital, Taipei 10630, Taiwan; m0587304.05g@g2.nctu.edu.tw (Y.-P.C.); aicragyang@gmail.com (Y.L.); aiqrsu@gmail.com (C.-Y.W.); 4Division of Pathology, Sijhih Cathay General Hospital, New Taipei City 22174, Taiwan; cyl1124@gmail.com; 5Department of Dermatology, Shin-Kong Wu Ho-Su Memorial Hospital, Taipei 11101, Taiwan; drkellytang@gmail.com; 6Department of Biochemistry, National Defense Medical Center, Taipei 11490, Taiwan; 7Department of Dermatology, College of Medicine, National Taiwan University, Taipei 10617, Taiwan

**Keywords:** interleukin-23, interleukin-17, psoriasis, salvianolic acid B, microemulsion, imiquimod, skin barrier

## Abstract

Psoriasis is a chronic inflammatory skin disorder with a pathogenesis involving the interleukin-23/interleukin-17 axis. Salvianolic acid B exerts several pharmacological effects, such as antioxidation, anti-inflammation, and antitumor effects. The anti-psoriatic effects of salvianolic acid B have not been reported. In this study, we aimed to determine the optimum vehicle for salvianolic acid B, investigate its therapeutic effect on psoriatic-like skin conditions, and explore its underlying mechanisms of action. BALB/c mice were administered topical imiquimod to induce psoriasis-like skin and were then randomly assigned to control, vehicle control, salvianolic acid B in vehicles, and 0.25% desoximetasone ointment treatment groups. Barrier function, cytokine expression, histology assessment, and disease severity were evaluated. The results showed that salvianolic acid B-containing microemulsion alleviated disease severity, reduced acanthosis, and inhibited interleukin-23/interleukin-17 (IL-23/IL-17) cytokines, epidermal proliferation, and increased skin hydration. Our study suggests that salvianolic acid B represents a possible new therapeutic drug for the treatment of psoriasis. In addition, such formulation could obtain high therapeutic efficacy in addition to providing sufficient hydration for dry skin.

## 1. Introduction

Psoriasis is a common chronic inflammatory skin disease, with histopathological features of epidermal hyperproliferation, abnormal keratinocyte differentiation, angiogenesis with blood vessel dilatation, and excess T helper 1 cell (Th-1) and T helper 17 cell (Th-17) lymphocyte infiltration [1,2,3]. It has been proposed that Th17 lymphocytes and interleukin-23 (IL-23) contribute to the psoriatic phenotype. IL-23, secreted by dermal dendritic cells, can induce Th17 lymphocyte differentiation with a subsequent release of proinflammatory cytokines on keratinocytes, leading to epidermal hyperplasia and parakeratosis [3,4].

Salvianolic acid B (Sal. B, C_36_H_30_O_6_, MW = 718.62) is one of the most abundant components of Danshen, which is the dried root of *Salvia miltiorrhiza*. In ancient Chinese herbal medicine, Danshen was used to treat angina pectoris, hyperlipidemia, and acute ischemic stroke due to its pharmacological effects, such as antioxidation, anti-inflammation, and antitumor effects [5,6,7,8,9]. Dermatology studies have reported that Sal. B had pro-angiogenesis, antiapoptosis, and antioxidative stress effects by stimulating autophagy that enhances the survival of skin flaps and wound healing [10,11,12]. Sal. B also provides skin photoprotection by inhibiting mitogen-activated protein kinase signaling pathways and activator protein-1 activation [13]. However, the effect of Sal. B on psoriasis has not yet been reported.

Potentially useful therapeutic compounds that are lipophilic (log P 1–3) and water soluble, with a molecular weight under 500 Dalton can successfully diffuse across the stratum corneum [14,15]. Sal. B is a hydrophilic compound, and its molecular weight is 718.62 Dalton [9]. These chemical properties might limit the efficacy of Sal. B due to insufficient skin penetration. To serve its therapeutic purposes, topical delivery of Sal. B would require assistance from other molecules to act as carrier vehicles or skin penetration enhancers in a controlled-release manner [14]. Using penetration enhancers is a standard approach to improving transdermal drug delivery by overcoming skin barrier resistance. Alcohols such as ethanol (EtOH), isopropanol, and n-propanol have been evaluated for their penetration-enhancing activity and drug solubility [16,17,18]. In addition, a skin irritation study has indicated that EtOH did not disrupt the skin barrier or exacerbate irritation even on irritated skin [19,20]. There is consensus that emollients and moisturizers containing humectants that hydrate the stratum corneum are a standard adjuvant therapy for managing inflammatory skin disorders such as psoriasis [21,22,23,24]. This study intends to determine the optimal carrier system for Sal. B, and then investigate the therapeutic effects of a Sal. B-containing formulation on an imiquimod (IMQ)-induced psoriatic-like dermatitis mouse model. The underlying mechanisms were also studied.

## 2. Materials and Methods

### 2.1. Materials

Salvianolic acid B, chloramphenicol, silicon oil AR200, triglyceride (TG), squalene, Tween 80, polyethylene glycol (PEG)-40 castor oil (RH-40), polyethylene glycol (PEG 400), 1,2-propylene glycol (1,2-PG), sorbitol, glycerol, and phosphate-buffered saline (PBS) tablet were purchased from Merck KGaA (Darmstadt, Germany). All other chemicals were of analytical grade.

### 2.2. Microemulsion Preparation

For determination of the existence zone of microemulsions (MEs), pseudo ternary phase diagrams were constructed using aqueous titration method. To construct pseudo ternary phase diagrams, the selected oil phase (silicon oil AR200:squalene:TG, 10:1:3) was mixed with a surfactant:cosurfactant mixture (RH40:Tween 80:PEG 400:1,2-PG, 10:1:1:1). The ratios of the mixture used for titration were 9:1, 8:2, 7:3, 6:4, 5:5, 4:6, 3:7, 2:8, and 1:9, and the mixture was titrated with PBS buffer (50% sorbitol and 15% glycerol) until it turned turbid. The volume of the water phase used was recorded.

#### 2.2.1. Formulation Selection

The formulations were then screened by the mouse skin irritation study. The formulation showing the least irritation confirmed by histological staining was selected for further studies.

#### 2.2.2. Stability Study

The selected formulation was subjected to thermodynamic stability tests including the centrifugation test and stress test (heating–cooling and freeze–thawing cycles). The selected formulation was centrifuged at 3500 rpm for 30 min to ensure physical stability. Stress test was carried out at 4 °C and 45 °C for 48 h each for a period of six cycles, followed by 25 °C and −21 °C for 48 h for three cycles. The samples were checked for phase separation. During the studies, the drug content was not determined [25].

### 2.3. Microemulsion Characterization

#### 2.3.1. Measurement of Droplet Size and Zeta Potential

Droplet size and zeta potential were measured by a Nanoparticle Analyzer (SZ-100, Horiba Ltd., Kyoto, Japan) at a scattering angle of 90° under ambient conditions.

#### 2.3.2. Electronic Conductivity

Electronic conductivity was measured by Eutech COND 6+ (Eutech Instruments Pte Ltd., Thermo Fisher Scientific, Singapore, Singapore) under ambient conditions.

#### 2.3.3. Viscosity

The viscosity was measured by a Visco-895 Viscometer (Atago Co., Inc., Tokyo, Japan) using spindle A3 RE-77106 at room temperature.

### 2.4. Animals

Male BALB/c mice (6–8 weeks old, National Laboratory Animal Center, Tainan, Taiwan) were housed under the standard laboratory conditions of controlled humidity (40%) and temperature (24 ± 2 °C), with a 12 h light/dark cycle. All animal experiments were conducted according to accepted standards of humane animal care, under protocols approved by the Institutional Animal Care and Use Committee (IACUC) of Cathay General Hospital (108-006, 19 December 2018). During the experimental period, each animal was housed in a separate cage to prevent activities between mice that could affect the measurements of skin barrier functions. In addition, each individual cage was provided with toys such as wooden bars to comply with IACUC regulations.

#### Skin Irritation and Study Design

For the skin irritation study, mice were randomly assigned to (a) control group: IMQ-induced for six consecutive days, followed by discontinuation of all treatment for five days, and to (b) selected formulation bases (A to E) group: IMQ-induced for six consecutive days, followed by the application of selected formulation bases (A to E) for five days. Mice were sacrificed on day 11 and mice skin was fixed in 10% formalin. The formulation showing the least irritation confirmed by histological staining was selected for further studies.

### 2.5. Preparation of Mice Skin for In-vitro Skin Permeation and Deposition Studies

Normal untreated mice were sacrificed after anesthesia. Full-thickness dorsal skin was excised and the hair on the dorsal area was removed using a depilatory cream (Nair, Church & Dwight Co., Inc., Ewing, NJ, USA). Subcutaneous tissue was surgically removed. The skin was washed with distilled water for immediate use [26]. For in-vitro skin permeation and deposition studies, mouse skin was randomly assigned to (a) formulation A group or (b) EtOH group.

### 2.6. In-vitro Skin Permeation Study

In-vitro permeation studies through mice skin were performed using a diffusion cell sampling system (Shishin Technology Co., Taipei, Taiwan). Skin samples were mounted onto the diffusion cells (area 0.985 cm^2^) equilibrated at 37 ± 0.5 °C for 30 min before study. Then, 200 μL (300 μg/mL) of selected Sal. B formulation was loaded into donor cells. Ethanol was added to the receiver buffer (EtOH:PBS buffer, pH 7.4 = 2:8) to increase the solubility of the Sal. B and to maintain sink condition [26]. The receiver fluid was stirred by a Teflon-coated mini magnetic bar at a speed of 600 rpm and equilibrated at 37 ± 0.5 °C for 6 h. Receiver samples were filtered using a 0.22 μm membrane filter and then underwent high-performance liquid chromatography (HPLC) analysis.

### 2.7. Permeation Data Analysis

The cumulative amount of Sal. B that permeated the skin (Q, μg/cm^2^) was plotted against a function of time (h). The skin flux (permeation rate) at a steady state (Js, μg/cm^2^/h) was calculated from the slope of the linear portion of the curve.

### 2.8. Skin Deposition Study

A skin deposition study was conducted to quantify the amount of Sal. B deposited in the skin. After 6 h of an in vitro permeation study, the skin surface was washed five times with ethanol:water (1:1) and then cut into small pieces and homogenized with acetonitrile to denature proteins. The homogenate, which contained chloramphenicol (0.1 mg/mL, as internal standard), was centrifuged at 13,200 rpm for 10 min at room temperature. The supernatant (20 μL) underwent HPLC analysis. The Sal. B extraction recovery rate from whole skin was 91.5 ± 11.6% at a Sal. B concentration of 0.1 μg/mL.

### 2.9. HPLC System

The analysis was performed by a Primaide 1110 pump, a Primaide 1410 UV detector, and a Primaide 1210 auto-sampler (Hitachi, Tokyo, Japan). A Mightysil RP-18 column, 4.6 × 250 mm, 5 μm (Kanto Chemical Co. Inc., Tokyo, Japan) was used. The mobile phase was methanol-Milli Q water (60:40, *v*/*v*, pH 3 adjusted by orthophosphoric acid), filtered through a 0.22 μm Millipore filter. The flow rate was 0.6 mL/min, and the sample injection volume was 20 μL. Detection was performed at a wavelength of 288 nm for Sal. B and chloramphenicol (internal standard) at room temperature and detected at a retention time of 6.0 ± 0.1 min and 8.3 ± 0.1 min in Sal. B and chloramphenicol, respectively. The Sal. B standard curve showed power regression at a concentration of 0.1 μg/mL to 10 μg/mL. The limit of detection (LOD) and limit of quantification (LOQ) of Sal. B were 0.1 μg/mL. The inter- and intra-day assay accuracy (% error) and precision (% coefficient of variation (CV)) were −2.6% to −13.2% and 0.8% to 5.1%, respectively, for Sal. B at a concentration of 0.1 μg /mL (Appendix A).

### 2.10. Barrier Function Study

For the barrier function study, the mice were randomly assigned to (a) control group: IMQ-induced only; (b) EtOH group: IMQ-induced plus EtOH treatment; (c) formulation A group: IMQ-induced plus formulation A base treatment; (d) Sal. B/formulation A group (Sal. B/formulation A): IMQ-induced plus 300 μg/mL of Sal. B in formulation A treatment; and (e) desoximetasone ointment (DXM) group: IMQ-induced plus Esperson (0.25% desoximetasone ointment (DXM) Sanofi, Handok Inc., Seoul, Korea) treatment as a positive control. After 3–4 h of IMQ application, 100 μL of Sal. B/formulation A, formulation A, or 60 mg DXM were applied once daily to the dorsal skin (Appendix A).

#### 2.10.1. Imiquimod-induced Psoriasis-like Skin Animal Model

Psoriasiform dermatitis was induced in the mice following a standard protocol [27,28,29]: 6–8-week-old male BALB/c mice were administered a daily topical application of 62.5 mg IMQ cream (5%) (Aldara, 3M Pharmaceuticals, Saint Paul, MN, USA) on the shaved dorsal skin for six consecutive days (daily dose of 3.125 mg of IMQ).

#### 2.10.2. Assessment of Barrier Function

Transepidermal water loss (TEWL), skin hydration, and skin erythema values were measured on the dorsal surface of the mice before the application of drugs on day 0, day 3, and day 6 using an MPA 2 system equipped with Tewameter TM300, Corneometer CM825, and Mexmeter MX18 probes (Courage and Khazaka, Köln, Germany).

### 2.11. Collection of Skin Specimens

Normal untreated mice were sacrificed 48 h after shaving of hair, and the full-thickness skin served as a negative control for inflammatory cytokines analysis. The mice for barrier function study were sacrificed on day 6 after the barrier function assessment. The full-thickness mouse skin was separated into two samples for histological staining and proteins were extracted.

### 2.12. Inflammatory Cytokine Protein Determination

Proteins were extracted from whole skin for the cytokine analysis (interleukin-17A (IL-17A), interleukin-17C (IL-17C), interleukin-17F (IL-17F), interleukin-22 (IL-22), interleukin-23 (IL-23), and tumor necrosis factor alpha (TNF-α)). Protein samples were then stored at −80 °C before analysis. IL-17C protein expression was analyzed using the mouse IL-17C kit (ABcam, Cambridge, MA, USA) and determined by a microplate reader (BioTek, Winooski, VT, USA) set to 450 nm wavelengths according to the manufacturer’s instructions. The other cytokines were determined by the LEGENDplex™ Kits (mouse inflammation panel, BioLegend, San Diego, CA, USA). The samples were incubated with labeled microbeads and each cytokine was determined by flow cytometry (Accuri C6, BD Biosciences, San Jose, CA, USA) according to the manufacturer’s instructions. 

### 2.13. Proliferation Cell Nuclear Antigen (PCNA) Protein Determination

Epidermal protein was extracted from epidermis prepared from full-thickness mouse skin and the PCNA concentrations were determined by a mouse PCNA ELISA quantization kit (Cell Biolabs Inc., San Diego, CA, USA) according to the manufacturer’s instructions.

### 2.14. Histological Staining

Skin samples were fixed in 10% formalin. After routine processing and embedding in paraffin, 5 μm-thick sample sections were cut, stained with hematoxylin and eosin, and examined under a light microscope (Olympus BX41, Tokyo, Japan).

### 2.15. Statistical Analysis

For skin permeation and deposition studies, Student’s *t*-test was used to determine statistical significance. For the other studies, one-way ANOVA was followed by the Scheffe post-hoc test to determine statistical significance. IBM SPSS statistics software Version 20 was used. *p* < 0.05 was considered statistically significant. Data were presented as mean ± SD in all tables. All bar figures were plotted as mean ± SD by Sigmaplot 10.0 software.

## 3. Results

### 3.1. Formulation Composition and Characterization of Selected Formulation

Figure 1A represents the chemical structure of Sal. B. The pseudo-ternary phase diagram of microemulsion consists of oil, surfactant, and water. Formulations A–E represent the studied formulations (Figure 1B). To select the suitable formulation, the skin irritation study was conducted by an IMQ-induced psoriasis-like dermatitis model. Figure 2A represents the design of the skin-irritation study. Formulation A treatment group showed similar pathological features as the control group with less severity index than all the other groups (Figure 2B). After the skin-irritation study, formulation A was then conducted by the thermodynamic stability test. Formulation A did not show phase separation and therefore formulation A was selected for further studies.

The composition and physicochemical properties of selected formulations are listed in Table 1. For formulation A, the selected oil phase was mixed with the surfactant and water phase by the titration 1:6:3. For the EtOH, 75% ethanol solution was selected. The droplet size, size distribution (polydispersity index, PI), zeta potential, viscosity, and electronic conductivity of formulation A were 696.2 ± 188.3 nm, 0.435 ± 0.004, –14.95 ± 0.64 mV, 3112.3 ± 5.8 cp, and 24.15 ± 0.07 μS/cm, respectively.

### 3.2. Skin Penetration Parameters and Deposition Amounts of Selected Formulations

Penetration parameters are listed in Table 2. Formulation A showed approximately 10-fold greater skin flux than EtOH (J_s_, *p* < 0.05, *t*-test), cumulative amount (Q, *p* < 0.05, *t*-test), and permeability coefficient (K_p_, *p* < 0.05, *t*-test). Ethanol showed a faster time to steady-state skin flux than formulation A (*p* < 0.05, *t*-test). The Sal. B skin deposition amounts for both formulations are shown in Figure 3. The skin deposition AUC_0-6_ (area under curve 0 to 6 h) of Sal. B/formulation A (15.45 ± 1.84 μg/g tissue h) was three-times higher than that of Sal.B/EtOH (5.95 ± 4.22 μg/g tissue h, *p* < 0.05, *t*-test). These penetration and deposition data indicate that formulation A might be a better topical delivery carrier for Sal. B than EtOH.

### 3.3. Formulation A Was Selected as the Topical Delivery Carrier for Sal. B

The study design is shown in Figure 4A. The barrier function measurement values at day 0 represent the normal baseline (TEWL: 7.39 ± 1.20 g/m^2^/h; skin hydration: 58.25 ± 5.84 arbitrary units (AU); erythema: 16.80 ± 1.74 AU). Compared with the value on day 0, the TEWL (40.79 ± 8.05 g/m^2^/h) and erythema (31.23 ± 4.17 AU) values in the control group increased significantly after applying IMQ-induced psoriasis-like dermatitis for six consecutive days. In contrast, the skin hydration value (3.69 ± 1.55 AU) of the control group decreased significantly on day 6 compared to that on day 0. These data indicate that the barrier function of mouse skin was compromised by the treatment with IMQ.

Compared with the control and EtOH groups, the formulation A group had lower TEWL values (14.80 ± 3.47 g/m^2^/h) but no significant differences on day 3 (*p* > 0.05, Figure 4B-1). However, the formulation A group showed markedly higher skin hydration values (45.59 ± 6.85 AU) than the other two groups on day 3 (both at *p* < 0.05, Figure 4C-1). There was no significant difference in erythema values between the three groups (*p* > 0.05, Figure 4D-1). Based on the skin penetration parameters and barrier function values, formulation A was selected as the topical delivery carrier of Sal. B for the in vivo anti-psoriasis mouse study.

### 3.4. Sal. B/Formulation A and 0.25% DXM Restored Barrier Function

Sal. B/formulation A (22.85 ± 6.69 g/m^2^/h) and DXM (20.49 ± 5.69 g/m^2^/h) treatment significantly restored the TEWL values compared to those of the control group (40.79 ± 8.05 g/m^2^/h) on day 6 (both at *p* < 0.05, Figure 4B-2). The Sal. B/formulation A (52.29 ± 6.91 AU) and DXM (35.49 ± 5.56 AU) groups showed higher hydration values than that of the control group (22.29 ± 4.82 AU, both at *p* < 0.05) on day 3. In addition, the Sal. B/formulation A group (13.94 ± 7.04 AU) showed higher skin hydration values than that of the control group (3.84 ± 1.60 AU) and DXM (6.10 ± 4.03 AU) groups on day 6 (*p* < 0.05, Figure 4C-2). No treatments affected the erythema on psoriasis-like mouse skin (all *p* > 0.05, Figure 4D-2). These results indicate that Sal.B/formulation A and DXM treatment ameliorated the IMQ-induced barrier disruptions.

### 3.5. Sal. B/Formulation A and DXM Improve Psoriasis-like Dermatitis

The morphology of all groups showed scaly, erythematous, and dry skin (Figure 5A). The histopathology results showed acanthosis, parakeratosis, tortuous capillary dilatation in the papillary dermis, and inflammatory cell infiltration in all groups (Figure 5B). However, the Sal. B/formulation A and DXM treatment groups showed fewer severe clinical and pathological features than any of the other groups (Figure 5A,B). 

### 3.6. Sal. B/Formulation A and DXM Inhibit IL-23, IL-17A, IL-17F, and IL-22 Protein Expression in Psoriasis-like Skin

The cytokine array results showed that all cytokine expression was significantly increased in the control group compared with the negative control group. Both Sal. B/formulation A and DXM treatment inhibited IL-17A, IL-17F, IL-22, and IL-23 protein expression (all *p* < 0.05), but not IL-17C (*p* > 0.05) and TNF-α (*p* > 0.05, Figure 5C).

### 3.7. Sal. B/Formulation A and DXM Inhibit Hyperproliferation of Keratinocytes

To confirm the antiproliferative effects of Sal. B on keratinocytes, we determined the protein expression of PCNA, a marker strictly associated with cell proliferation. The ELISA results showed that the expression of PCNA was significantly decreased in the Sal. B/formulation A (*p* < 0.05) and DXM (*p* < 0.05) groups (Figure 5D).

## 4. Discussion

Crude coal tar and its derivatives remain a safe and highly effective option for the treatment of psoriasis vulgaris. Crude coal tar contains thousands of ingredients. Heterocyclic aromatic compounds, such as benzofuran, are typical constituents of coal tars [30,31]. Benzofuran (C_8_H_6_O, MW = 118.13) is a heterocyclic aromatic compound consisting of fused benzene and furan rings. Due to the wide range of biological activities and potential applications, benzofuran and its derivatives have been developed as pharmacological agents for several aspects such as antioxidant, antitumor, antiplatelet, antimalarial, anti-inflammatory, antidepressant, and anticonvulsant properties [32,33]. Psoralen (C_11_H_6_O_3_, MW = 186.16), a compound structurally related to benzofuran, and its derivatives, 8-methoxsalen and 5-methoxsalen, have photosensitizing activity and are used orally and topically in conjunction with ultraviolet irradiation for the therapy of psoriasis and vitiligo [34]. Sal. B is one of the main water-soluble active ingredients responsible for the pharmacologic effects of Danshen and Sal. B is also a compound structurally related to benzofuran. Studies have reported the efficacy of Sal. B in various dermatology fields [10,11,12,13]. There have been fewer studies assessing Sal. B and its effects on psoriasis. The hydrophilicity and high molecular weight of Sal. B could limit its cutaneous penetration; to overcome this limitation, we developed a microemulsion formulation as its vehicle.

The IL-23/IL-17 axis has been reported to be a critical regulator for psoriasis and psoriatic arthritis. Studies have reported that IMQ-induced psoriasis-like skin elicited either protein or mRNA expression of IL-17, IL- 23, IL-1β, IL-6, TNF-α, and interferon-γ(INF-γ) in mouse skin [3,35,36,37] and successful anti-psoriasis treatment inhibited the above-mentioned cytokine expression [35,36,37]. Our study demonstrated a significant increase in IL-23/IL-17 cytokine protein expression after six consecutive days of IMQ application. Subsequent treatment with both Sal. B/formulation A and DXM markedly inhibited protein expression of IL-17A, IL-17F, IL-22, and IL-23, but not TNF-α or IL-17C. In addition, the ELISA results demonstrated that the epidermal proliferation marker PCNA was significantly reduced by topical treatment with Sal. B/formulation A and DXM. These results suggest that Sal. B had an anti-psoriasis effect equivalent to that of corticosteroids and that it could be developed as an alternative therapeutic drug for psoriasis treatment. 

The goals of psoriasis treatment are mainly symptom control and improved quality of life. The choice of treatment options for psoriasis include corticosteroids, vitamin D3 analogues, disease-modifying anti-rheumatic drugs (DMARDs), such as methotrexate, acitretin, and cyclosporine, and newly developed biological-targeted agents such as ustekinumab, secukinumab, and ixekizumab [38,39]. For patients with mild to moderate psoriasis, both expensive biologics and systemic treatment that can cause serious side effects were not reasonable treatment choices, thus making it worthwhile to develop effective and economical topical therapeutic options. In addition, the irritation sensation limits the usage of DMARDs as with conventional topical agents such as vitamin D3 analogues [40]. Compared with currently available treatment for psoriasis, our Sal. B-containing microemulsion is a phytopharmaceutical combined nanoformulation that works as effectively as potent topical corticosteroids. The formulation of the vehicle can also minimize skin irritation, which may be seen in some second-line topical treatments, including salicylic acid and tar. Therefore, Sal. B/ microemulsion could be an alternative topical treatment for psoriasis.

It has been proposed that improving hydration of the stratum corneum and diminishing inflammation using topical emollients could facilitate the prevention and mitigation of certain inflammation-associated chronic disorders such as psoriasis [21,22,23,24,41,42,43,44]. When used alone or as an adjunct, essential topical therapy can restore and protect skin barrier function, prolong the interval between psoriasis flare-ups, and enhance the effects of pharmaceutical therapy [45,46]. Osborne et al. reported that the successful topical delivery of hydrophilic drugs must have sufficient mobility of water within the microemulsion formulation and sufficient percutaneous transport of water across the skin barrier [47]. Our skin penetration results showed that formulation A enhanced the skin deposition of the hydrophilic active ingredient. It is possible that formulation A might also enhance the skin penetration of aquatic ingredients, including water and humectants, into the deep skin to provide sufficient skin hydration for a low-grade inflammatory condition. Thus, formulation A could also be considered an emollient carrier.

Microemulsion is defined as a transparent, thermodynamically stable, isotropic mixture that entraps emulsion droplets of micron to nano-size particles in oil/water/surfactants. Microemulsion is recognized as an efficient delivery method in terms of systemic or percutaneous absorption [48,49,50]. The selected formulation, formulation A, is a clear solution that is entrapped in oil/water/surfactants. Its droplet size and zeta potential are around 700 nm and −15 mV, respectively. Zeta potential is an essential index of stability of the suspension. The greater the absolute zeta potential, the higher the stability of colloidal dispersions [51]. In addition, our stability results show that formulation A was thermodynamically stable following centrifugation and stress tests; thus, formulation A represents a stability microemulsion.

The value of electronic conductivity can determine the nature of the continuous phase of the microemulsion. When water forms the continuous phase of the microemulsion, it shows an elevated conductivity. In contrast, low conductivity represents a continuous oil phase of the microemulsion [52]. Subongkot and Ngawhirunpat suggested that formulations with a conductivity ranging from 10.3 to 52.5 μS/cm can be defined as bicontinuous microemulsions [15,52]. The conductivity of formulation A was about 25 μS/cm and thus can be characterized as a bicontinuous microemulsion delivery system.

The nature of the continuous phase of the microemulsion may influence skin penetration. When applying the same hydrophilic drug caffeine in *o*/*w*, *w*/*o*, and bicontinuous microemulsions across excised pigskin, the transdermal flux of caffeine is in the following order: *w*/*o* < bicontinuous < *o*/*w* microemulsion. The *o*/*w* microemulsion provides the greatest permeation within 24 h [53]. In contrast, Zhang and Michniak-Kohn reported that the dermal delivery enhancement ratio of the hydrophobic drug, clotrimazole, also presented with the same order [54]. Thus, a drug in the *o*/*w* microemulsion may provide the highest amount of drug delivery with the bicontinuous system working as an intermediate delivery system [15,53,54].

Apart from the characteristics of the formulations, to achieve an enhanced topical drug delivery, it is sometimes necessary to perform physical or biomolecular structural alterations to the stratum corneum (SC) using suitable techniques, specific chemical agents or drug carriers [55]. Although changing or damaging the skin structure either by chemical or physical methods can increase the permeability of drugs via reduced lag time and increased permeation rates, such manipulations may affect the barrier integrity [56]. 

In our study, formulation A represented a stable bicontinuous microemulsion delivery system and presented as the least irritation formulation, and its microemulsion properties, skin flux, cumulative amount, and permeability coefficient were significantly higher than those of EtOH. The skin deposition AUC_0-6_ of Sal. B in formulation A was also markedly higher than that of EtOH. As a result, formulation A was selected as the optimal topical delivery system for Sal. B in this study. 

A previous study reported that TEWL values correlate well with the psoriasis area and severity index (PASI) scores, which are the most common and widely used tool for assessing the severity of psoriasis [57]. In pathological skin sites, there is an elevated TEWL value and a low corneum water content due to the disrupted stratum corneum [58,59]. TEWL value is a quantitative and non-invasive indicator that evaluates the barrier function of the stratum corneum of diseased skin [60,61,62]. The values of TEWL and hydration are well correlated with the visual assessment of skin barrier function [59]. Our results have shown that the TEWL and erythema values in the control group increased significantly after IMQ-induced psoriasis-like dermatitis. In contrast, the skin hydration value of the control group decreased significantly after IMQ-induced psoriasis-like dermatitis. Sal. B/formulation A and DXM treatment significantly restored the TEWL values compared to those of the control group on day 6. Notably, the Sal. B/formulation A group showed higher skin hydration values than those of the control and DXM groups. The barrier function results suggest the therapeutic efficacy of Sal. B/formulation A and DXM. 

In summary, our study suggests that a Sal. B-containing microemulsion formulation could be a good candidate for topical anti-psoriasis treatment by reducing the inflammatory response by downregulating the IL-23/IL-17 axis, inhibiting the abnormal proliferation of keratinocytes, and providing sufficient hydration for dry skin.

## 5. Conclusions

In this study, Sal. B showed non-inferior therapeutic effects compared with desoximetasone for IMQ-induced psoriasis treatment. The results revealed that simultaneous inhibition of IL-17, IL-23, and abnormal epidermal proliferation by desoximetasone and Sal. B contributed to improving the barrier function and the therapeutic effect of IMQ-induced psoriasis-like dermatitis in mice. In addition, the microemulsion design could enhance the skin penetration of Sal. B, which had a high molecular weight and was hydrophilic. Our study shows that such a formulation could obtain high therapeutic efficacy in addition to providing sufficient hydration for dry skin.

## Figures and Tables

**Figure 1 pharmaceutics-12-00457-f001:**
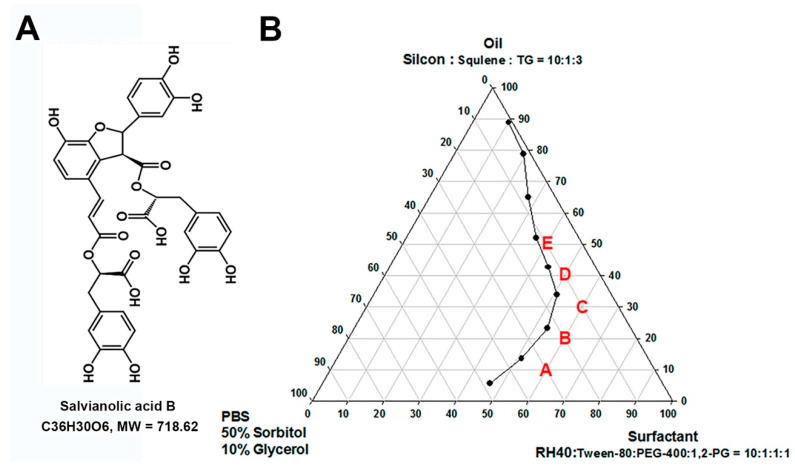
Active compound and candidate topical carriers. (**A**) Chemical structure of salvianolic acid B (Sal. B). (**B**) The pseudo-ternary phase diagram of microemulsion consisting of oil (silicon oil AR200:squalene:triglyceride (TG), 10:1:3), surfactant (RH40:Tween 80:polyethylene glycol (PEG) 400:1,2-PG, 10:1:1:1), and water (50% sorbitol and 15% glycerol in phosphate-buffered saline). Formulations A–E represent the studied formulations. The composition (oil:surfactant:water) by the titration of formulation A was 1:6:3, whereas the titration of formulation B was 2:6:2, formulation C was 3:6:1, formulation D was 4:5:1, and formulation E was 5:4:1.

**Figure 2 pharmaceutics-12-00457-f002:**
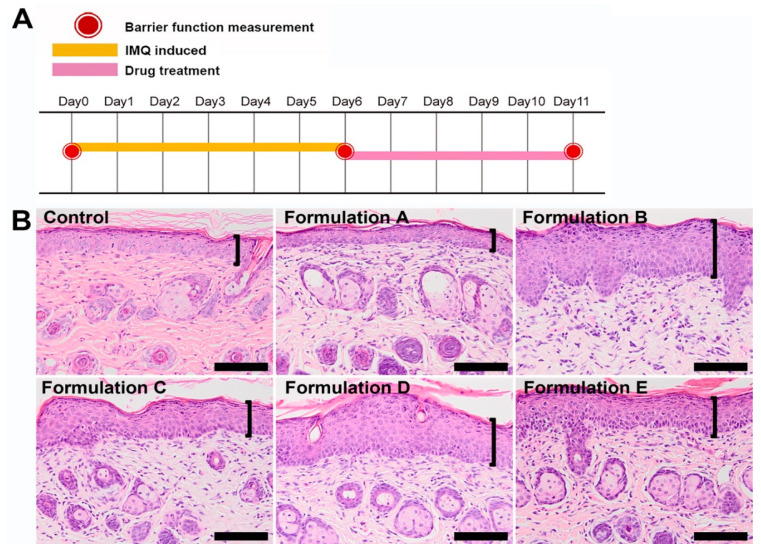
Selection of the topical delivery carrier for Sal. B. (**A**) To select the suitable formulation for inflammatory skin disease, the skin-irritation study was conducted by imiquimod (IMQ)-induced psoriasis-like dermatitis model. (**B**) The histopathological results showed that the formulation A treatment group had similar pathological features as the control group with less skin irritation than all the other groups; therefore, formulation A was selected for further studies. Scale bar = 100 μm. Brackets indicate thickness of epidermis.

**Figure 3 pharmaceutics-12-00457-f003:**
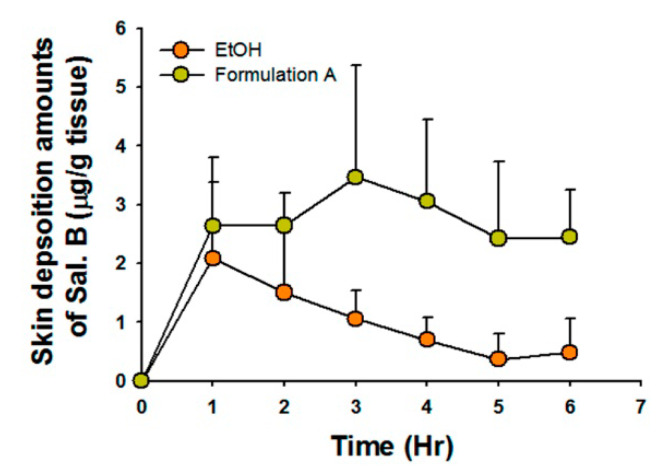
Skin deposition of Sal. B in different vehicles. The skin deposition AUC_0-6_ (area under curve 0 to 6 h) of Sal. B/formulation A was significantly higher than that of Sal. B/EtOH.

**Figure 4 pharmaceutics-12-00457-f004:**
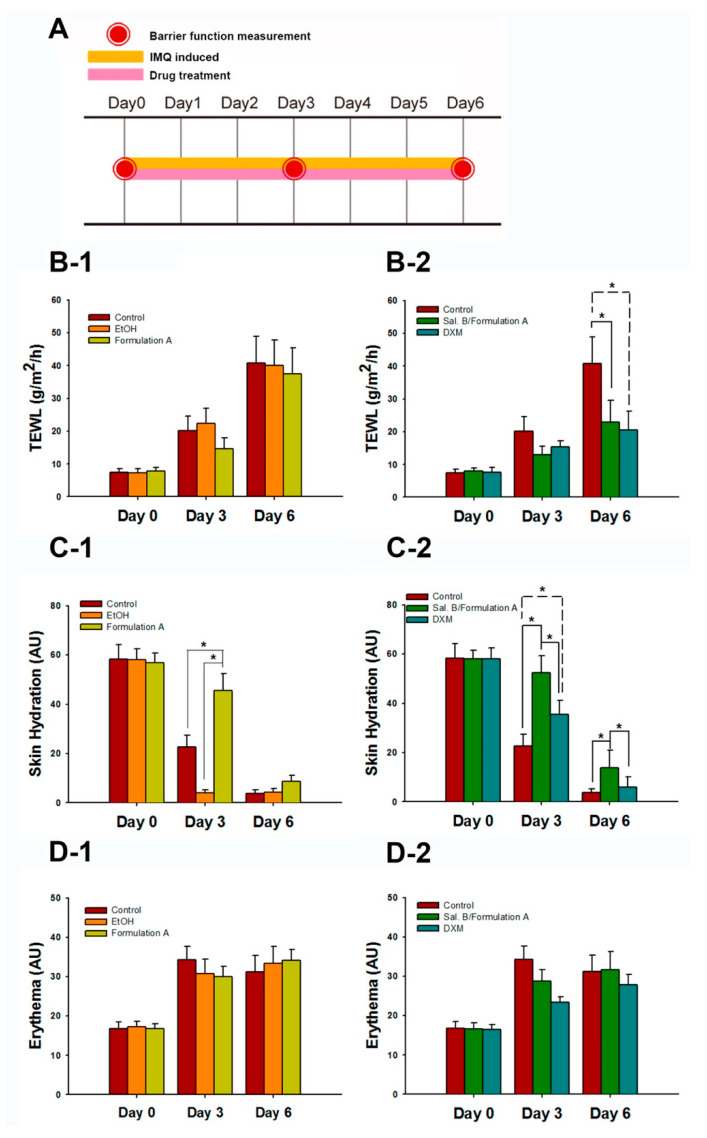
Sal. B/formulation A and desoximetasone ointment (DXM) restored the barrier function. (**A**) The study design. After three to four hours of IMQ application, mice received 100 μL Sal. B in Formulation A, Formulation A, or 60 mg DXM on the dorsal skin once daily. Barrier functions were measured on the dorsal skin before the application of drugs on day 0, day 3, and day 6. (**B-1**) Transepidermal water loss (TEWL), (**C-1**) skin hydration, and (**D-1**) DXM represented the change of barrier function for the in vivo topical carrier selection study. (**B-2**) TEWL, (**C-2**) skin hydration, and (**D-2**) erythema represented the change in barrier function for the anti-psoriasis treatment study. (mean ± SD, *n* = 4); * *p* < 0.05; one-way ANOVA (Scheffe).

**Figure 5 pharmaceutics-12-00457-f005:**
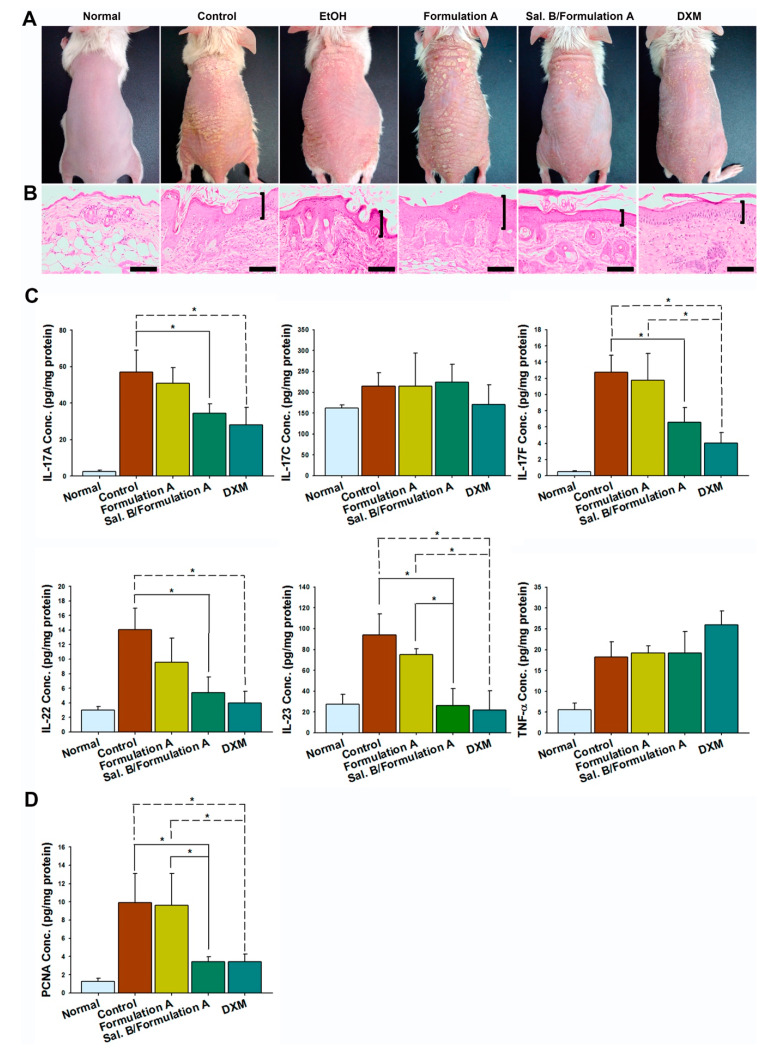
Sal. B/formulation A and DXM improve psoriasis-like dermatitis. (**A**) The morphology of all groups showed scales, erythema, and dry skin. (**B**) Histopathology by hematoxylin and eosin (H&E) staining showed epidermal acanthosis, parakeratosis, tortuous capillary dilatation in papillary dermis, and inflammatory cell infiltration in all groups. The Sal. B/formulation A and DXM treatment groups showed fewer severe clinical and pathological features than any of the other groups. Scale bar = 100 μm. (**C**) Sal. B/formulation A and DXM inhibited interleukin-17A (IL-17A), interleukin-17F (IL-17F), interleukin-22 (IL-22), and interleukin-23 (IL-23) protein expression but not interleukin-17C (IL-17C) and tumor necrosis factor alpha (TNF-α). (**D**) Sal. B/formulation A and DXM inhibited proliferation cell nuclear antigen (PCNA) protein expression (mean ± SD, *n* = 4); * *p* < 0.05; one-way ANOVA (Scheffe).

**Table 1 pharmaceutics-12-00457-t001:** Physicochemical properties of selected formulation.

Properties (Unit)	Formulation A
Composition	O:S:W = 1:6:3
Droplet size (nm)	696.2 ± 188.3
Size distribution (PI)	0.435 ± 0.004
Zeta potential (mV)	−14.95 ± 0.64
Viscosity (cP)	3112.3 ± 5.8
Electronic conductivity (μS/cm)	24.15 ± 0.07

Sal. B, salvianolic acid B; O, oil phase (silicon oil AR200:squalene:TG, 10:1:3); TG, triglyceride; S, surfactant phase (RH40:Tween 80:PEG 400:1,2-PG, 10:1:1:1); RH40, PEG-40 castor oil; PEG 400, polyethylene glycol; 1,2-PG, 1,2-propylene glycol; W, water phase (50% sorbitol and 15% glycerol in phosphate buffered saline); EtOH, ethanol; -, un-tested; PI, polydispersity index.

**Table 2 pharmaceutics-12-00457-t002:** Penetration parameters of Sal. B in different vehicles through normal murine skin after 6 h (mean ± SD, *n* = 3).

Parameters (Unit)	Formulation A	EtOH
J_s_ (ng/cm^2^ h)	574.44 ± 280.00 *	52.66 ± 8.09
t_lag_ (hr)	1.78 ± 0.32	0.82 ± 0.14 *
Q (ng/cm^2^)	2411.23 ± 877.94 *	305.14 ± 65.05
K_p_ (×10^−3^ cm/h)	1.91 ± 0.93 *	0.18 ± 0.03

Sal. B; Salvianolic acid B; EtOH, ethanol; J_s_, steady-state flux; t_lag_, lag time; Q, cumulative amount, where Q = concentration × volume of diffusion cell/area; K_p_, permeability coefficient, where K_p_ = J_s_/drug concentration in donor cell; * *p* < 0.05, Student’s *t*-test.

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
