# Peer review of "Salvianolic Acid B in Microemulsion Formulation Provided Sufficient Hydration for Dry Skin and Ameliorated the Severity of Imiquimod-induced Psoriasis-like Dermatitis in Mice"

_pharmaceutics, 2020, doi:10.3390/pharmaceutics12050457_

Round 1

Reviewer 1 Report

The manuscript "Salvianolic acid B in microemulsion formulation provided sufficient hydration for dry skin and ameliorated the severity of imiquimod-induced psoriasis-like dermatitis in mice"  covered with a variety of studies, formulation development, ex vivo drug permeation and deposition, and animal studies to show tolerability of the formulations.

The author did not explain about the stability studies of the formulation and comments, either additional stabilities may helpful to improve the quality of work

Author Response

Reviewer  #1

The manuscript "Salvianolic acid B in microemulsion formulation provided sufficient hydration for dry skin and ameliorated the severity of imiquimod-induced psoriasis-like dermatitis in mice"  covered with a variety of studies, formulation development, ex vivo drug permeation and deposition, and animal studies to show tolerability of the formulations.

The author did not explain about the stability studies of the formulation and comments, either additional stabilities may helpful to improve the quality of work

Response: We appreciate your suggestions and have conducted the stability studies, followed by centrifugation test and stress tests which included heating–cooling and freeze–thawing cycles tests (Ryu et al., Pharmaceutics 2020, 12, doi:10.3390/pharmaceutics12040332). Formulation A did not show phase separation. It represented as a stable formulation. We highlighted the methods in Material and Methods paragraph 2.2.2 (Line 91—97 on page 2-3), the results in Results paragraph-Formulation composition and characterization of selected formulation (Line 237-240, p.6), and in Discussion (Line 415-423 on page 17).

Reviewer 2 Report

There are various grammatical errors throughout the manuscript that must be addressed. Although I mention a few of those throughout my review, authors are highly recommended to edit the manuscript carefully.

Major: The authors need to describe the currently available antipsoriatic options and the novelty of Sal. B. compared with them. What are the advantages of Sal. B. compared with the currently available options. 

Minor:

1- Throughout the introduction section, there are abbreviations (Th-1, Th-17, IL-23, ...) that were not expanded. 

It is recommended to review the manuscript and write the full term on the first appearance of each term.

2- In the materials section, the purchase source of Sigma is mentioned as "France". Please confirm if this is right.

3- The verb tenses throughout the manuscript shall be revised.

For example, in the results section, the first sentence is written in past presence whereas the next sentence is in present form.

"Figure 1A represented the chemical structure of Sal. B. The pseudo-ternary phase diagram of microemulsion consists of oil, surfactant, and water." > revise "represented" to "represents"

There are more grammatical errors throughout the manuscript that shall be addressed. Please review the manuscript for grammatical errors once again. 

4- In figure 1, the section "A", "B", and "C" are too small that is almost impossible to see what the authors are trying to show.

- Inside section "D" there are subsections labeled as "A" to "E", which overlaps with the main sections of figure 1. It can be revised to "Formulation A" etc....

- Please use arrows to highlight the areas of interest in section "D".

- Section "E" is too small to read.

- The title of the figure legend can be revised as follows: "Formulation A was selected as the topical delivery carrier for Sal. B." > "Selection of the topical delivery carrier for Sal. B."

- In figure 1's legend, the authors mentioned "Chemical structures of Sal. B." whereas there is only a single chemical structure shown.

- Also "(D) The histopathological results showed that the 214 formulation A treatment group treatment group showed similar pathological" shall be revised.

5-  In figure 2, the figure texts in B-1 ~ D-2 are too small. I recommend the authors to modify the arrangement of the sections to improve the readability of the figures. 

- The font size of section labels "A", "B-1", etc... need to be matched with the rest of the figures.

6- In figure 3, the section labels "A", "B", etc. are recommended to be placed on the left top side of each section to improve clarity.

- Please use arrows to highlight the areas of interest in section "B".

- The texts in section "C" and "D" need to be enlarged.

7- The last column of table 1 can be avoided. 

8- Table 3 can be avoided and written as text in the manuscript. 

Author Response

Reviewer  #2

There are various grammatical errors throughout the manuscript that must be addressed. Although I mention a few of those throughout my review, authors are highly recommended to edit the manuscript carefully.

Major: The authors need to describe the currently available antipsoriatic options and the novelty of Sal. B. compared with them. What are the advantages of Sal. B. compared with the currently available options.

Response: The goals of psoriasis treatment are mainly symptom control and improved quality of life. The choice of treatment options for psoriasis include corticosteroids, vitamin D3 analogues, disease-modifying anti-rheumatic drugs (DMARDs) such as methotrexate, acitretin, and cyclosporine, and newly developed biological-targeted agents such as ustekinumab, secukinumab, and ixekizumab (Francesco et al., Expert Rev Clin Pharmacol 2020, 10.1080/17512433.2020.1759415, doi:10.1080/17512433.2020.1759415, Rendon et al., Int J Mol Sci 2019, 20, doi:10.3390/ijms20061475). For patients with mild to moderate psoriasis, both expensive biologics and systemic treatment that can cause serious side effects were not reasonable treatment choices, thus making it worthwhile to develop effective and economical topical therapeutic options. In addition, the irritation sensation limits the usage of DMARDs as with conventional topical agents such as vitamin D3 analogues (Campanati at al., Expert Opin Drug Saf 2020, 19, 439-448, doi:10.1080/14740338.2020.1740204). Compared with currently available treatment for psoriasis, our Sal. B-containing microemulsion is a phytopharmaceutical combined nanoformulation that works as effectively as potent topical corticosteroids. The formulation of the vehicle can also minimize skin irritation, which may be seen in some second-line topical treatment, including salicylic acid and tar. Therefore, Sal B/ microemulsion could be an alternative topical treatment for psoriasis. (Line 390-402 on page 16).

Minor:

1- Throughout the introduction section, there are abbreviations (Th-1, Th-17, IL-23, ...) that were not expanded.

It is recommended to review the manuscript and write the full term on the first appearance of each term.

Response: Thank you for your comments. We have made corrections as suggested : T helper 1 cell (Th-1), T helper 17 cell (Th-17) (Line 42 on page 1), interleukin-23 (IL-23) (Line 43 on page 1), interleukin-17A (IL-17A), interleukin-17C (IL-17C), interleukin-17F (IL-17F), interleukin-22 (IL-22), and tumor necrosis factor alpha (TNF-α) (Line 207-209 on page 5).

2- In the materials section, the purchase source of Sigma is mentioned as "France".

Please confirm if this is right.

Response: We have corrected it as Merck KGaA (Darmstadt, Germany) (Line 76-77 on page 2) since Sigma Aldrich had merged with Merck. Thank you for your comment.

3- The verb tenses throughout the manuscript shall be revised.

For example, in the results section, the first sentence is written in past presence whereas the next sentence is in present form.

There are more grammatical errors throughout the manuscript that shall be addressed.

Please review the manuscript for grammatical errors once again. 

Response: The revised manuscript has been edited by the MDPI service (English editing ID: english-18495).

4- In figure 1, the section "A", "B", and "C" are too small that is almost impossible to see what the authors are trying to show.

- Inside section "D" there are subsections labeled as "A" to "E", which overlaps with the main sections of figure 1. It can be revised to "Formulation A" etc....

- Please use arrows to highlight the areas of interest in section "D".

- Section "E" is too small to read.

- The title of the figure legend can be revised as follows: "Formulation A was selected as the topical delivery carrier for Sal. B." > "Selection of the topical delivery carrier for Sal. B."

- In figure 1's legend, the authors mentioned "Chemical structures of Sal. B." whereas there is only a single chemical structure shown.

- Also "(D) The histopathological results showed that the 214 formulation A treatment group treatment group showed similar pathological" shall be revised.

Response: Thank you for your suggestions.  The modifications were as follow:

- We separated A and B in Figure 1, C and D in Figure 2, E in Figure 3.

- We corrected Figure 1A's legend from “Chemical structure”s” of Sal B” to “Chemical structure of Sal. B." (Line 243 on page 6)

- We rephrased all the labels accordingly in Figure 2. (Line 257 on page 8)

- The grammatical error of Figure 2's legend was revised as “The histopathological results showed that the Formulation A-treatment group had similar pathological features as the control group with less skin irritation than all the other groups." (Line 260-261 on page 8)

- We used brackets instead of arrows to highlight the changes of the epidermal thickness in Figure 2, Section "B". (Line 257 on page 8)

- The title of Figure 2 was revised as “Selection of the topical delivery carrier for Sal. B”. (Line 258, page 8)

5-  In figure 2, the figure texts in B-1 ~ D-2 are too small. I recommend the authors to modify the arrangement of the sections to improve the readability of the figures.

- The font size of section labels "A", "B-1", etc... need to be matched with the rest of the figures.

Response: Thank you for your suggestions. Modifications of Figure 4 were as follow:

- 14 font size were used and the presentations were rearranged from B-1 to D-2 sections to improve the readability of the figures. (Line 315 on page 12)

- The font size of section labels "A", "B" was kept at 24 font size throughout all figures. 

6- In figure 3, the section labels "A", "B", etc. are recommended to be placed on the left top side of each section to improve clarity.

- Please use arrows to highlight the areas of interest in section "B".

- The texts in section "C" and "D" need to be enlarged.

Response: Thank you for your suggestions. Our modifications were as follow:

- The section labels "A", "B", etc. were placed on the upper left side of each section to improve the clarity of Figure 5. (Line 352 on page 15)

- We used brackets instead of arrows to highlight the changes of the epidermal thickness in Figure 5, Section "B". (Line 352 on page 15)

- In section C&D, we enlarged the font size of the figure text from 10 to 14. (Line 352 on page 15)

7- The last column of table 1 can be avoided.

Response: The last column of table 1 was deleted as suggested. (Line 270 on page 8)

8- Table 3 can be avoided and written as text in the manuscript.

Response: Table 3 was deleted and was written as text in the revised manuscript. (Line 281-282 on page 9)

Reviewer 3 Report

The manuscript is an interesting presentation of the activity of Salvianolic acid B for the treatment of psoriasis. Anyway, even if the experimental design seems correct, the manuscript is written in a confusing way and the presentation of the work needs major revisions before being published. Moreover, the English language needs to be revised by a native speaker.

-Introduction lines 64-65: about the ethanol activity on the skin, a more recent reference describing the topical use of the solvent has to be cited (i.e. International Journal of Cosmetic Science, 2017, 39, 188–196)

-paragraph 2.3.1: in the results it is reported also the zeta potential measurement. Please add it in this section.

-lines 104-119 paragraph 2.4.1: in my opinion this paragraph is inconsistent in this position. I suggest to revise it and to move the description of each study design in the relevant paragraph.

- paragraph 2.4.2: in my opinion it is better to move this paragraph before paragraph 2.10.

-paragraph 2.5 Was the skin used immediately after the excision or was it frozen prior to use? if yes, how long?  

-lines 134-135: specify the composition of the receiving phase

-paragraph 2.6: replace ex-vivo with in-vitro

-paragraph 2.7.:in my opinion it is not necessary to report the formulas, as they are know to most of the public. Anyway, it should be useful to describe the meaning of the different parameters, including lag time and permeability coefficient, since you calculated and reported them in the table.

-paragraph 2.9: insert LOD and LOQ values for Sal B.

-Figure 1: there are so many results in this figure that is totally confusing. I suggest to separate A and B in one picture and C and D in another one. E should be alone in the paragraph of the permeation/penetration study. Moreover, in the caption are discussed also some results, but they must be reported in a form that suits the caption of a figure and then they will be discussed in detail in the relevant paragraphs. 

- Table 1: the column regarding EtOH can be removed, it is not relevant.

Author Response

Reviewer  #3

The manuscript is an interesting presentation of the activity of Salvianolic acid B for the treatment of psoriasis. Anyway, even if the experimental design seems correct, the manuscript is written in a confusing way and the presentation of the work needs major revisions before being published. Moreover, the English language needs to be revised by a native speaker.

Response: Thank you for your suggestions. The manuscript was edited and  revised by MDPI service (English editing ID: english-18495).

-Introduction lines 64-65: about the ethanol activity on the skin, a more recent reference describing the topical use of the solvent has to be cited (i.e. International Journal of Cosmetic Science, 2017, 39, 188–196)

Response: The reference was cited as reference #18 as suggested:  

Alcohols such as ethanol (EtOH), isopropanol, and n-propanol have been evaluated for their penetration-enhancing activity and drug solubility [16-18]. (Line 63-64 on page 2)

-paragraph 2.3.1: in the results it is reported also the zeta potential measurement. Please add it in this section.

Response: Droplet size and zeta potential were measured using Nanoparticle Analyzer (SZ-100, Horiba Ltd., Kyoto, Japan). We corrected the paragraph title from "Measurement of droplet size and electronic conductivity “to "Measurement of droplet size and zeta potential". (Line 100-102, p.3, pharmaceutics-780654-revised)

-lines 104-119 paragraph 2.4.1: in my opinion this paragraph is inconsistent in this position. I suggest to revise it and to move the description of each study design in the relevant paragraph.

Response: We revised the manuscript and corrected paragraph 2.4.1 as "Skin irritation study design".  (Line 118-124 on page 3)

We moved the "skin permeation and deposition studies" section into paragraph 2.5: "Preparation of mice skin for in-vitro skin permeation and deposition studies". (Line 139-144 on page 4)

We moved the "barrier function study" section as paragraph 2.10. (Line 182-189 on page 5)

- paragraph 2.4.2: in my opinion it is better to move this paragraph before paragraph 2.10.

Response:  We moved the section "Imiquimod induced psoriasis-like skin animal model from paragraph 2.4.2 to paragraph 2.10.1 (Line 190-194 on p age 5)

-paragraph 2.5 Was the skin used immediately after the excision or was it frozen prior to use? if yes, how long?

Response: The mouse skin was used immediately after the excision. Our experience showed that the mouse skin would be severely damaged after freezing and could not be used for skin permeation study. We revised the manuscript as follow:

The skin was washed with distilled water for immediate use. (Line 143 on page 4)

-lines 134-135: specify the composition of the receiving phase

Response: We added the composition of the receiving phase as follow:

The receiver fluid (EtOH:PBS buffer, pH 7.4 = 2:8) was stirred by a Teflon-coated mini magnetic bar at a speed of 600 rpm and equilibrated at 37 ± 0.5℃ for 6 hours. (Line 151 on page 4)

-paragraph 2.6: replace ex-vivo with in-vitro

Response: We replaced the term “ex-vivo" with "in-vitro" in Material and Method paragraph 2.5(Line 139 and 143 on page 4), 2.6(Line 146 and 147 on page 4), and 2.8(Line 164 on page 4).

-paragraph 2.7.:in my opinion it is not necessary to report the formulas, as they are know to most of the public. Anyway, it should be useful to describe the meaning of the different parameters, including lag time and permeability coefficient, since you calculated and reported them in the table.

Response: We deleted the formulas in paragraph 2.7. We listed these two formulas as word text in table 2:

Q, cumulative amount, where Q = concentration x volume of diffusion cell / area;

Kp, permeability coefficient, where Kp =  Js / drug concentration in donor cell; (Line 288-289 on page 9)

-paragraph 2.9: insert LOD and LOQ values for Sal B.

Response:  The LOD and LOQ values for Sal B in paragraph 2.9 were inserted as follow:

The limited of detection (LOD) and limited of quantification (LOQ) values of Sal. B were 0.1 μg/mL. (Line 179-180 on page 5)

-Figure 1: there are so many results in this figure that is totally confusing. I suggest to separate A and B in one picture and C and D in another one. E should be alone in the paragraph of the permeation/penetration study. Moreover, in the caption are discussed also some results, but they must be reported in a form that suits the caption of a figure and then they will be discussed in detail in the relevant paragraphs.

Response: Modifications were as follow:

- We separated A and B in Figure 1, C and D in Figure 2, kept E alone in Figure 3. (Figure 1, Line 242 on page 7;  Figure 2, Line 257 on  page 8; Figure 3, Line 294 on page 9)

- We reported the results in compliance to the caption of a figure and also discussed in detail in the relevant paragraphs. (Figure 1, Line 362-378 on page 16; Figure 2, Line 439-446 on page 17; Figure 3, Line 439-449 on page 17)

- Table 1: the column regarding EtOH can be removed, it is not relevant.

Response: The last column of table 1 was deleted. (Line 270 on page 8)

Round 2

Reviewer 2 Report

The authors have addressed the major issues raised.

Figure labels (A), (B), etc. shall be matched with the figure size. For example, in figure 1 the labels are too large for the figure.  

Also, supplementary figure 1  (HPLC data) is not clear.

Author Response

Reviewer  #2

The authors have addressed the major issues raised.

Figure labels (A), (B), etc. shall be matched with the figure size. For example, in figure 1 the labels are too large for the figure. 

Response: Thank you for your suggestions. The font size of figure 1 labels "A", "B" was change as 20 font size to match with the figure 1 size. (Line 243 on page 7)

Also, supplementary figure 1  (HPLC data) is not clear.

Response: Thank you for your suggestions. We reformatted the supplementary figure 1 (HPLC data) to improve the clarity. (Line 478 on page 19)

Reviewer 3 Report

All the correction have been performed and the manuscript has been improved. The manuscript can be accepted in its present form.

Author Response

Reviewer 3

All the correction have been performed and the manuscript has been improved. The manuscript can be accepted in its present form.

Response: We appreciate your suggestions.